# Investigation of Bemethyl Biotransformation Pathways by Combination of LC–MS/HRMS and In Silico Methods

**DOI:** 10.3390/ijms22169021

**Published:** 2021-08-21

**Authors:** Daria A. Belinskaia, Elena I. Savelieva, Georgy V. Karakashev, Olga I. Orlova, Mikhail A. Leninskii, Nataliia S. Khlebnikova, Natalia N. Shestakova, Alexandra R. Kiskina

**Affiliations:** 1Sechenov Institute of Evolutionary Physiology and Biochemistry, Russian Academy of Sciences, Pr. Torez 44, 194223 St. Petersburg, Russia; esavelieva59@mail.ru (E.I.S.); karakashev58@mail.ru (G.V.K.); orlolga@yandex.ru (O.I.O.); m.leninskii@yandex.ru (M.A.L.); nskhlebnikova@mail.ru (N.S.K.); n_shestakova@list.ru (N.N.S.); kiskina-aleksandra@rambler.ru (A.R.K.); 2Research Institute of Hygiene, Occupational Pathology and Human Ecology, Federal Medical Biological Agency, Kapitolovo Station, G/P Kuzmolovsky, Vsevolozhsky District, Leningrad Region, 188663 Kuzmolovsky, Russia

**Keywords:** bemethyl, 2-(ethylthio)benzimidazole, actoprotector, doping, rat urine, metabolite, LC–MS/HRMS, in silico metabolism prediction, glutathione S-transferase, molecular docking

## Abstract

Bemethyl is an actoprotector, an antihypoxant, and a moderate psychostimulant. Even though the therapeutic effectiveness of bemethyl is well documented, there is a gap in knowledge regarding its metabolic products and their quantitative and qualitative characteristics. Since 2018, bemethyl is included to the Monitoring Program of the World Anti-Doping Agency, which highlights the challenge of identifying its urinary metabolites. The objective of the study was to investigate the biotransformation pathways of bemethyl using a combination of liquid chromatography-high-resolution mass spectrometry and in silico studies. Metabolites were analyzed in a 24 h rat urine collected after oral administration of bemethyl at a single dose of 330 mg/kg. The urine samples were prepared for analysis by a procedure developed in the present work and analyzed by high performance liquid chromatography–tandem mass spectrometry. For the first time, nine metabolites of bemethyl with six molecular formulas were identified in rat urine. The most abundant metabolite was a benzimidazole–acetylcysteine conjugate; this biotransformation pathway is associated with the detoxification of xenobiotics. The BioTransformer and GLORY computational tools were used to predict bemethyl metabolites in silico. The molecular docking of bemethyl and its derivatives to the binding site of glutathione S-transferase has revealed the mechanism of bemethyl conjugation with glutathione. The findings will help to understand the pharmacokinetics and pharmacodynamics of actoprotectors and to improve antihypoxant and adaptogenic therapy.

## 1. Introduction

The benzimidazole core is present in a large number of drugs with a broad spectrum of biological activity [1]. The most known benzimidazole drugs produced in Russia are dibazole and etomerzole. Although these drugs are widely used in clinical practice, their metabolism has not yet been studied. Bemethyl (commercial names bemaktor and metaprot) is an actoprotector, an antihypoxant, and a moderate psychostimulant, which, like dibazole, contains a benzimidazole core, but unlike dibazole, contains a 2-ethylthio instead of 2-benzyl substituent. The mechanism of action of bemethyl involves the activation of the cell genome and a positive modulating effect on protein synthesis in the body [2].

It causes a rapidly developing enhancement of ribonucleic acid (RNA) and protein synthesis in different organs and tissues. Bemethyl is used as an anti-asthenic agent in the therapy of neuroses, organic brain lesions of the traumatic and infectious genesis with the prevalence of asthenic symptoms, and myopathies. It is also used in the case of ischemic heart disease, radiation sickness, and infectious diseases, as well as in cardiac surgery. Bemethyl was recommended as antihypoxant and adaptogen for the correction of conditions associated with extreme physical and mental stress [3].

Even though the therapeutic effectiveness of bemethyl is well documented [4,5], as well as its pharmacokinetics [6,7], there is a significant gap in knowledge regarding its metabolism products and their quantitative and qualitative characteristics. Most of the published studies are focused on medical issues, while the chemical processes underlying the pharmacological effects of bemethyl remain unexplored. In particular, Zarubina and Shabanov [6] reported that the unchanged drug was detected in the plasma of volunteers after taking a single dose at concentrations above 4 ng/mL for no longer than 10 h, but little was mentioned about the forms of drug elimination from the body.

Boĭko et al. [7] studied the pharmacokinetics of bemethyl in rats by measuring the concentrations of unchanged drug in blood plasma by gas chromatography (GC). Two routes of administration were compared: intravenous and oral. With both routes of administration, biexponential kinetics of the elimination of bemethyl from the body reached a maximum concentration within 1 h. Studying the urinary excretion of bemethyl, the authors of the cited work found that the unchanged form of bemethyl comprised as little as 0.56% of the total excreted drug.

Kwiatkowska et al. [8] noted rapid metabolism of this compound but considered the only urinary metabolite bemethyl glucuronide. The authors detected bemethyl in the urine samples of volunteers 58 h after administration, while its glucuronide could be detected in up to 78 h. Zarubina and Shabanov [6] described two other possible metabolites of bemethyl, specifically its oxidized forms 2-(ethylsulfinyl)- and 2-(ethylsulfonyl)benzimidazoles. Until now no other possible urinary metabolites of bemethyl have been mentioned in the literature. The molecular and structural formulas of bemethyl and some most closely related benzimidazole drugs are presented in Table 1.

The biotransformation pathways of bemethyl in the body are important to study for understanding the nature of its antioxidant and protective properties, and such study is of considerable interest. In our previous work, we reported the chemical formulas of six compounds that were obtained by the examining of urine samples of rats treated with bemethyl and suggested that these compounds could be considered as possible bemethyl metabolites [9]. In that work, we used a different experimental model to study bemethyl metabolism: a course of intragastric administration at a daily dose of 25 mg/kg. For 3 days, bemethyl was administered to rats intragastrically at a daily dose of 25 mg/kg, followed by daily urine sampling for 26 days. On the 26th day, the mean concentration of unchanged bemethyl in rat urine was at the limit of LC-MS/HRMS detection 5.2 ± 1.5 ng/mL.

In the present work, using a single dose of 330 mg/kg intragastrically, we have performed, for the first time, an in-depth study of the composition of the products of bemethyl metabolism in rats by a combination of liquid chromatography-high-resolution mass spectrometry (LC–MS/HRMS) and in silico methods, and made assumptions about the metabolism pathways. The conditions of LC-MS/HRMS analysis were basically the same as in [9].

## 2. Results and Discussion

To identify the urinary metabolites of bemethyl in rats and gain a general insight into its biotransformation in the body, we made use of the results of the LC–MS/HRMS analysis, literature data on the metabolic transformations of other drugs of the benzimidazole series (Table 1), in silico metabolism prediction and molecular docking studies. With all the great potential of LC–MS/HRMS to identify xenobiotic metabolites in multicomponent biomatrices, the xenobiotic biotransformation pathways are impossible to establish without additional information. Such information can be gained from molecular modeling analysis, as well as known data on the metabolism of the closest structural analogs of the studied xenobiotics. Molecular modeling allows the prediction of possible biotransformation pathways with an estimate of the probability of each possible pathway. Published data on the biotransformation of structural analogs provide the necessary supporting information. Thus, the biotransformation pathway of bemethyl to sulfoxide and sulfone, with a stable hydroxylated form arising only with sulfoxide but not with sulfone, reveals a complete analogy with the biotransformation of triclabendazole. The ability of pantoprazole and omeprazole to form a conjugate with *N*-acetylcysteine suggests the same biotransformation pathway of bemethyl.

The extremely high dose of bemethyl was chosen with two goals in mind: (1) to detect as many major metabolites as possible and preserve the original component ratio of the sample, because concentration inevitably entails a change in its composition and (2) to identify metabolites associated with detoxification. The animal experimental model used would be unsuitable for pharmacokinetic studies and only serves to identify metabolites.

Enzymatic hydrolysis was not used for the direct determination of conjugates. Extraction procedures will be developed for streaming target analysis. During the primary identification phase, direct urine analysis seems to be optimal.

### 2.1. Tandem Mass Spectral Identification of the Main Urinary Metabolites of Bemethyl

Analytes were identified using Thermo Fisher Scientific MetWorks 1.3 software developed for the search for drug metabolite identification. The molecular formulas are generated with the characteristic ions chosen by an analysis of the primary tandem mass spectral information. Raw LC–MS/HRMS data are presented in the Appendix A.

The advantage of the MetWorks software is that it allows isolation of the signals of xenobiotics against the background of interfering matrix signals in the HPLC–MS*^n^* chromatograms and spectra. The MetWorks software automatically generates ion mass chromatograms for the metabolism products of the test compounds, which might be expected to form along a selected metabolic pathway, the MS*^n^* spectra and retention times (RT) of these products, as well as mass numbers (*m/z*) of characteristic signals. At the first stage, the software generates the primary table of accurate masses, from which the duplicate data arising from the contribution of adducts and isotopic interferences, as well as bio matrix signals are removed. Then, with the account for the characteristic isotopic distributions and MS/MS*^n^* spectral–structural correlations, the product ions and mass fragments, characteristic of typical metabolic transformations, are established. The principal tandem mass spectral and chromatographic characteristics of bemethyl and its metabolites are listed in Table 2.

The fact that we established structural formulas for a number of metabolites did not allow an unambiguous conclusion. The mass chromatograms of metabolites M1–M3 each contain two peaks with different retention times, which suggest the presence of two tautomers with the same molecular formula. Thus, the tandem mass spectra of peaks with retention time (RT) 9.40 min (M1а) and 11.01 min (M1b) are identical and display the following signals: 151.0324 (100%); 119.0604 (5%); 118.0531 (14%); 93.0581 (29%); 92.0495 (4%).

The identical tandem mass spectra of metabolites M1а and M1b provide evidence showing that metabolite M1 (2-thiobenzimidazole) exists in the urine as a mixture of two tautomers. The thione–thiol tautomerism of 2-thiobenzimidazole is well understood. As is known, the thione form is more stable [10] and prevails in aqueous solutions at рН < 10 [11]. The known structures of the tautomers of M1 and the possible charge localization are presented in Figure 1.

The structures were proposed on the basis of the MetWorks data and confirmed by the calculation of exact masses using the Mass Frontier software. As seen from Table 2, the product ions of both tautomers have the same molecular formulas. The area ratio of the peaks in the mass chromatogram shows that the concentration of the tautomer with RT 9.40 min is higher by a factor of 60 than the concentration of the tautomer with RT 11.01 min.

The mass chromatogram (*m/z* 195.0593 → 167.0278) of the second metabolite of bemethyl (M2), too, displays two peaks corresponding to ions with the same molecular formula [C_9_H_10_N_2_OS+H]^+^. The retention times of these peaks are 9.13 min (M2а) and 11.42 min (M2b), and their area ratio is 40:1, respectively. The tandem mass spectra of M2а (195.0587 (100%); 167.0279 (94%); 166.0195 (12%); 134.0480 (5%)) and M2b (195.0587 (5%); 167.0279 (100%); 149.0173 (42%)) differ from each other in ion compositions and peak intensities, implying that the molecular formula C_9_H_10_N_2_OS corresponds to two different structural formulas. For example, the ion peak at *m/z* 166.0195 [C_7_H_6_N_2_OS]^+^ in the mass spectrum of the peak of M2а (RT 9.13 min) presumably corresponds to a sulfoxide structure (Figure 2A). The ion peak at *m/z* 149.0173 [C_7_H_5_N_2_S]^+^ in the mass spectrum of the peak of M2b (RT 11.42 min) is assignable to a mercaptane structure (Figure 2B). It is well known (since 1990s) that, in metabolic studies, hydroxylation and S-oxidation products can be differentiated by hydrogen/deuterium exchange (HDX) experiments. The investigation of bemethyl metabolites by HDX technique is one of our future research challenges.

Figure 3 represents the proposed structural formulas of metabolites M2. The localization of the OH group may be different, as well as charge/proton could be located at any heteroatom position (here and below).

Metabolite 3 (M3), too, gives two peaks in the mass chromatogram (RT 9.26 min and 10.51 min) of the product ions formed from the precursor ion with *m/z* 227.0486 (Figure 4). The area ratio of the two peaks is 1:10, and their tandem mass spectra are identical (227.0486 (52%); 199.0176 (18%); 181.0066 (100%); 153.0117 (7%); 135.0553 (13%)), and, therefore, we cannot assign a concrete structure to one or the other peak. Two variants of the proposed structures of the isomers of M3 are presented in Table 2, and the proposed structures of the product ions, generated by the Mass Frontier software, are shown in Figure 4.

The mass spectrum of metabolite M4 contains signals at *m/z* 275.0155 (100%), 276.0191 (10%), and 277.0114 (10%). The theoretical mass spectrum reconstructed on the basis of the molecular formula of metabolite M4 is identical to that which is experimental. The *m/z* 275.0154 and 277.0114 peak intensity ratio shows that the molecule contains two sulfur atoms (according to the isotope distribution of sulfur).

Metabolite M4 is formed by the hydroxylation and subsequent sulfonation of bemethyl. The proposed structural formula of the precursor ion of M4 [C_9_H_10_N_2_O_4_S_2_+H]^+^ is presented in Table 2, and the proposed structures of the product ions are given in Figure 5.

The molecular formula corresponding to the precursor ion of metabolite M5 is [C_12_H_13_N_3_O_3_S+H]^+^. This metabolite was not identified by the MetWorks software, because none of the metabolic pathways entered in the program involve benzimidazole ring opening.

Two compounds can have this molecular formula: a conjugate of bemethyl with acetylcysteine (M5a) or a conjugate of bemethyl with glutamic acid (M5b). The proposed structures of the precursor ions are given in Table 2, and the structures of the product ions are depicted in Figure 6.

As will be shown below, on the basis of the published data on the biotransformation of bemethyl analogs, M5 with a high probability should be considered an *N*-acetylcysteine conjugate of bemethyl.

Metabolite M6 is a conjugate of hydroxyl bemethyl with glucuronic acid. Its mass spectrum contains the following signals: 371.0908 (0.2%), 195.0587 (100%), and 167.0274 (10%). The structure of the precursor ion is given in Table 2, and the proposed structures of the product ions are depicted in Figure 7.

Presumably, metabolite M6 is a conjugate of hydroxylated bemethyl with glucuronic acid. It is important to note that the only bemethyl metabolite that were detected in human urine by Kwiatkowska et al. [8] is bemethyl glucuronide (the *m/z* values of the precursor and most characteristic product ions are 355.00 and 179.00, respectively). The respective values for metabolite M6 are 371.091 and 195.0587, which provides evidence showing that this metabolite is, indeed, hydroxylated bemethyl glucuronide. We also detected the conjugate of bemethyl with glucuronic acid (*m/z* 355.0958) in rat urine, but its peak was weaker by 1–2 orders of magnitude compared to the peak of hydroxylated bemethyl glucuronide (M6). In view of the fact that the molecular formulas of M1, M2, and M3 each corresponded to two products with different retention times, it is safe to state that we detected nine bemethyl metabolites in rat urine. These compounds were all detected in the urine samples of rats administered with bemethyl and not detected in blank urine samples. Thus, in the absence of the corresponding reference samples, we can consider the detected metabolites as biomarkers of bemethyl, but their structure can only be assigned on an assumptive level.

Reasonable assumptions concerning the metabolism of bemethyl can be deduced from considering the known directions of biotransformation of other compounds of the benzimidazole series, as well as from in silico analysis.

### 2.2. Prognosis of the Metabolic Pathways of Bemethyl on the Basis of the Known Metabolism Pathways of Its Analogs

The structures of the metabolites formed as a result of «atypical» biotransformation reactions, such as ring cleavage or, vice versa, ring closure, are difficult to assess exclusively from tandem mass spectral data. The structure of unknown metabolites can sometimes be established by analogy with the metabolites of known structural analogs of the drug in hand. As mentioned above, bemethyl does not have close structural analogs, and, therefore, the benzimidazole derivatives in Table 1 may be useful for predicting some biotransformation pathways of this drug but not its overall metabolism.

A number of drugs with diverse biological activities have been developed on the basis of benzimidazole. Among the great variety of benzimidazole drugs mentioned in the review of Akhtar et al. [12], the antihelmintic drug triclabendazole (TCB) is the only containing an alkylthiol terminal group. Therefore, we considered just this drug as a structural analog of bemethyl.

The metabolism of TCB [6-chloro-5-(2,3-dichlorophenoxy)-2-(methylthio)- 1H-benzimidazole)] was studied not only in helminths, but also in warm-blooded animals [13]. The metabolism of this drug in sheep was found to involve its oxidation to sulfoxide and sulfone, as well as hydroxylation. It was also noted that the intestinal microflora of sheep is able to partially reduce the oxidized forms. The in vitro experiments with the microsomal fraction of the sheep liver showed that the biotransformation of the TCB occurs via consecutive formation of sulfoxide, sulfone, and hydroxy sulfone.

Mottier et al. [14] described five metabolites of TCB: triclabendazole sulfoxide, triclabendazole sulfone, hydroxy-triclabendazole, hydroxy-triclabendazole sulfoxide and hydroxy-triclabendazole sulfone. The same authors isolated TCB and all the five its metabolites from sheep bile [15]. They showed in their in vitro experiments that triclabendazole sulfoxide (active metabolite) is formed at a much higher rate than triclabendazole sulfone. Later, Cañas-Müller et al. [16] established that triclabendazole sulfoxide is an active metabolite, while triclabendazole sulfone is inactive. It can be suggested that the biotransformation of bemethyl, too, involves sulfoxide and sulfone formation. Therewith, the free hydroxylated form has been detected only for sulfoxide M4 (Figure 6). TCB is hydroxylated in the 2,3-dichlorophenoxy fragment. In view of the absence of such fragment in bemethyl, the analogy with TCB cannot help in establishing its hydroxylation site.

The closest analog of bemethyl among the drugs developed and produced in Russia is afobazole [5-ethoxy-2-[(2-(morpholino)ethylthio]benzimidazole dihydrochloride). The structural formula of afobazole is presented in Table 1. Bochkov et al. [17] described only two metabolites of afobazole, which are formed by the oxidation of the morpholine ring and its de-ethylation, followed by hydroxylation of the benzimidazole ring. Seredenin et al. [18] studied, in detail, the metabolism of afobazole in rats. By HPLC–MS analysis, the authors detected 17 metabolites of afobazole and made use of independent synthesis to establish the structure of six of them. One of the two main metabolites of afobazole is formed by the hydrolysis of the 5-ethoxy substituent in the benzimidazole ring. Bemethyl metabolite M6 is likely to be formed by this mechanism. The elimination of the morpholine fragment from afobazole is accompanied by cyclization via the addition of the ethylthio fragment to one of the nitrogen atoms of the benzimidazole ring. We did not detect a similar tricyclic derivative among the urinary metabolites of bemethyl, which is not surprising, because Seredenin et al. [18] mentioned that they detected the cyclization product in rat plasma, but not in urine and feces. The common feature of the metabolism of bemethyl and afobazole is that they both involve the oxidation of sulfur to sulfoxide and, further, to sulfone. This information is not new, because such a biotransformation pathway of bemethyl was already described by Kurpiakova et al. [19]. Furthermore, the same biotransformation pathway was also unambiguously confirmed for TCB [14,15,16].

The absolute conclusion that follows from the results of all known studies of the metabolism of bemethyl analogs is that the benzimidazole core remains intact.

The structure of the most abundant bemethyl metabolite M5 proved to be difficult to establish. As mentioned above, MetWorks offered no options, and similar biotransformation products have not previously been found among the metabolites of bemethyl analogs. As a possible hypothesis, we decided to consider the conjugation of bemethyl with glutathione. In doing so, we based this on the biotransformation of the proton pump inhibitor pantoprazole (Table 1), which, even though it contains a benzimidazole fragment, might seem at first glance to be structurally dissimilar to bemethyl. Zhong et al. [20] detected, among the urinary metabolites of pantoprazole, conjugates with both the pyridine and benzimidazole fragments of its molecule. In the latter case glutathione attacks the С2 position of the benzimidazole ring (marked with an asterisk in Table 1). This process is predominantly catalyzed by glutathione *S*-transferase (GST). A series of biotransformation reactions then occur by the classical mechanism of mercapturic acid formation, leading, in the end, to a stable adduct of benzimidazole with *N*-acetylcysteine.

Weidolf et al. [21] observed the same biotransformation reaction in the urine of rats after the administration of 400 mmol/kg of a mixture of [^3^H]- and [^14^C]-labeled omeprazole preparations. The main biotransformation pathways of omeprazole involved its dissociation into two parts: [^3^H]-labeled pyridine and [^14^C]-labeled benzimidazole. The benzimidazole fragment was identified as a conjugate with *N*-acetylcysteine. Such a biotransformation pathway is characteristic of many xenobiotics, but is better studied in toxicology than in pharmacology, as one of the ways of detoxification of xenobiotics and their elimination from the body [22].

By analogy with pantoprazole and omeprazole, we suggest that, in the case of bemethyl, glutathione, too, attacks the benzimidazole ring at the C2 position to form a conjugate of 2-mercaptobenzimidazole (metabolite M1) with glutathione (M1–SG). Most probably, like with omeprazole [20], the oxidized form of bemethyl, rather than bemethyl itself, undergoes conjugation; however, we have applied molecular docking method to check this assumption (see Section 2.4).

### 2.3. In Silico Prediction of the Metabolism of Bemethyl

To study the molecular mechanisms of bemethyl biotransformation pathways and explain the appearance of the metabolites detected in rat urine by the HPCL–MC/MC method, we have applied in silico tools.

The search for phase I bemethyl biotransformation products (cytochrome P450-dependent monooxygenase–catalyzed oxidation) was performed by means of the BioTransformer open access metabolism prediction software tool [23] and GLORY software tools [24]. GLORY predicts the sites of metabolism (SoM) using a Fast Metabolizer (FAME) algorithm [25], and then combines it with CYP-mediated reactions from the reaction rule set that was specifically created for GLORY. FAME is a machine learning model for the prediction of sites of metabolism for drug-like and other xenobiotic compounds [25]. Based on the calculated SoMs, GLORY gives the score to the predicted metabolites, indicating the probability of their formation: the higher the score, the higher the probability of the metabolite formation. The subsequent transformations of bemethyl (phase II) were suggested on the basis of the published evidence on the metabolism of related compounds. The detailed information about the GLORY procedure and its results is presented in Appendix A.

Based on our HPLC–MS/MS and in silico data, as well as on the data published in the literature, the biotransformation pathways of bemethyl can be proposed (Figure 8).

Bemethyl oxidation pathways and probabilities of oxidations products formation are presented in Appendix A. According to calculations using GLORY software, the most probable first-step oxidation products of bemethyl are compounds M2a, M2b, and Т1 (scores 4.20, 2.49, and 2.11, respectively) (Figure 8 and Appendix A). Compounds M2a and M2b are variants of metabolite M2 detected by mass spectrometry (Table 2). In its turn, compound Т2 is the most probable oxidation product of metabolite M2a (Appendix A), and compound M3 is formed by oxidation of Т2 (Appendix A). Compound M3 is one of the proposed metabolites M3 detected by mass spectrometry (Table 2).

Since the sulfur atom is more reactive that the benzimidazole carbon atoms (Appendix A), the triple oxidation of benzimidazole (metabolite M3a in Table 2) is less probable than the formation of structure M3b. There is one way of M3a formation:

Bemethyl → compound **3** (metabolite M2b) → compound **16** → compound **29** (metabolite M3a).

GLORY scores for these transformations are 2.49, 2.15 and 1.81.

For metabolite M3b there are three pathways:

Bemethyl → compound **2** (M2a) → compound **9** (T2) → compound **24** (M3b);

Bemethyl → compound **2** (M2a) → compound **10** → compound **24** (M3b);

Bemethyl → compound **3** (metabolite M2b) compound **10** → compound **24** (M3b).

For theses pathways, the GLORY scores are: 4.20, 4.14, 2.52; 4.20, 2.54, 2.52; 2.49, 3.87, 3.87.

If one deals with the scores as with probabilities, then the total score is 9.69 for M3a and 107.99 for M3b. Therefore, metabolite M3 is primarily present in urine as sulfone, which is consistent with our previous findings [9].

Furthermore, using the online service BioTransformer, we determined enzymes involved in the mentioned oxidation reactions: for example, the formation of metabolite M2a from bemethyl involves CYP1A2, CYP2A6, CYP2B6, CYP2C9, and CYP3A4 cytochrome P450 subtypes, metabolite M2b is formed with the participation of CYP1A2, CYP2A6, CYP2B6, CYP2C9, CYP2C19, and CYP3A4, compound Т2 is formed with the participation of CYP2C9 and CYP3A4, while CYP1A2, CYP2C9, CYP2C19, and CYP3A4 mediate the formation of metabolite M3 (Figure 8, Appendix A).

According to the GLORY calculations, compound Т1 is the third most probable oxidation product of bemethyl (Appendix A). However, BioTransformer did not predict the formation of this compound, and, therefore, we still do not have enough evidence to suggest the type of the cytochrome involved in this reaction. Compound Т1 can undergo oxidative dealkylation by a mechanism analogous to the mechanism of oxidative N-dealkylation [26]. The dealkylation product is metabolite M1 (Figure 8), which was identified by mass spectrometry. This metabolite, similar to other related compounds [10,27], can exist in two tautomeric forms: thiol and thione.

Zhong et al. [20] reported the metabolites of pantoprazole that are structurally related to bemethyl metabolite M2a (bemethyl sulfoxide). Both compounds can be described by the general formula R1–S(O)–R2. In the case of pantoprazole, R1 is 6-(difluoromethoxy)-1*H*-benzimidazole and R2 is (3,4-dimethoxypyridin-2-yl)methyl. The R1 and R2 in bemethyl are benzimidazole and ethyl, respectively. According to Zhong et al. [20], glutathione «breaks» the pantoprazole molecule into two fragments: conjugates of R1 and R2 with glutathione. We suggested that, by analogy with pantoprazole, one of the products of the reaction of metabolite M2a with glutathione is compound T3. The mechanism of this reaction is still unclear. Campbell et al. showed that molinate sulfoxide can undergo the conjugation with glutathione both enzymatically and spontaneously [28]. Gunnarsdottir and Elfarra showed that the conjugation of *cis*-3-(9*H*-purin-6-ylthio)acrylic acid (PTA, similar to bemethyl by its structure) with GSH in vitro occurred non-enzymatically only [29]. We have applied molecular docking studies to unravel the molecular mechanism of bemethyl enzymatic conjugation with GSH (see Section 2.4). Intermediate T3 transforms into metabolite M5, which was identified by mass spectrometry, via a sequence of enzymatic reactions involving γ-glutamyltransferase (GGT), cysteine–glycine dipeptidase (CGDP), and *N*-acetyltransferase (NAT).

Various compounds containing phenolic groups are eliminated from the body as sulfate conjugates. The reaction occurs in several steps. The first stage gives rise to an active form of sulfate, specifically 3-phosphoadenazine-*S*-phosphosulfate (PAPS). The transfer of the sulfo group to an acceptor molecule is catalyzed by a sulfotransferase enzyme (SULT) [30]. Thus, most likely, metabolite M4 is formed by the sulfation of the OH group in metabolite M2b (Figure 8).

Metabolite M6 evidently results from the conjugation of compound M2b with glucuronic acid via the OH group (Figure 8). As is known, the conjugation of xenobiotics with glucuronic acid is catalyzed by uridine 5’-diphospho-glucuronosyltransferase (UGT), the substrate of which is uridine-diphosphate-glucuronic acid (UDP-Gluc).

M5 and M4, and, to a lesser extent, M2 formed by aromatic hydroxylation, can be considered as the main metabolites. The content of glucuronide (M6) is relatively low.

### 2.4. Conjugation of Bemethyl and Its Oxidation Products with Glutathione According to Molecular Modeling Data

To determine the possible pathways for conjugation of bemethyl and its oxidation products with glutathione, and to explain the appearance of metabolite M5, we have applied molecular modeling methods. The enzymatic conjugation is catalyzed by GST [31]. In the GST superfamily, three subfamilies of GST isoforms are distinguished: cytosolic, mitochondrial and microsomal; the cytosolic isoforms are responsible for approximately 90% of the GST activity in the cell. Seventeen cytosolic GST isoforms in mammals are grouped into seven classes (α, μ, π, θ, ζ, ω, σ), which can be divided into two subgroups by the mechanism of catalysis. The first subgroup consists of the Y-GST isoforms (α, μ, π, and σ classes), in which the glutathione molecule is activated by tyrosine residue. The second one consists of S/C-GST isoforms activating GSH with the help of serine (φ, τ, θ, and ζ classes) or cysteine (β and ω classes) [32].

To analyze the possibility of enzymatic conjugation of bemethyl and its derivatives with glutathione, we have performed molecular docking of bemethyl, bemethyl sulfoxide (BEM-SO) and bemethyl sulfone (BEM-SO2) into the ligand-binding site of rat glutathione S-transferase (rGST) (Figure 9). Alpha 1 isoform (rGSTA1-1) with catalytic Tyr9 was chosen as the GST model for molecular docking procedures. The structure of the complexes obtained by molecular docking was optimized by the method of energy minimization (see “Materials and Methods” section for details). As mentioned above, glutathione attacks the benzimidazole ring at the C2 position to form a conjugate with glutathione. Therefore, in the complex of GST with GSH and bemethyl (or its derivatives), the distance between the sulfur atom of glutathione and the C2 atom of the ligand (distS-C2) should not exceed 4 Å in order nucleophilic attack to occur and a new covalent bond to form.

The superposition of bemethyl and its derivatives docked inside rGSTA1-1 with S-benzyl-glutathione inside human glutathione S-transferase alpha 1 (structure code 1GUH [33]) showed that the benzimidazole rings of the ligands bind in a similar configuration to the benzene ring of S-benzyl glutathione (Figure 9A). According to the data obtained, in the complex of GST with bemethyl, the distS-C2 value is equal to 5.72 Å (Figure 9B), which makes a nucleophilic substitution reaction unlikely. This result is in agreement with the data of Gunnarsdottir and Elfarra [29]. As was mentioned in Section 2.3, the authors studied the conjugation of PTA, similar to bemethyl by its structure, with GSH in vitro and showed that adding of various GST isoforms (including GSTA1-1) to the reaction mixture did not change the rate of accumulation of *S*-(9*H*-purin-6-yl)glutathione, which is the PTA−GSH conjugate.

In the case of bemethyl sulfoxide and bemethyl sulfone, the values of distS-C2 are 3.65 and 3.83 Å, respectively (Figure 9C,D), which is sufficient for nucleophilic attack and the formation of a new covalent bond between glutathione and atom C2. The side chain of Arg15 plays the key role in the binding of BEM-SO and BEM-SO_2_. It forms a hydrogen bond with the oxygen atoms of BEM-SO and BEM-SO_2_, stabilizing the position of the ligands in the productive conformation. The important role of Arg15 in the catalytic mechanism of GSTA1-1has been noted in a number of papers [34,35].

Based on the data obtained, we suppose that glutathione transferase is involved in the formation of metabolite M5, and the oxidation of the sulfur atom is necessary for this reaction to proceed. We believe that both sulfide and sulfone can be the substrates for phase II metabolism by cytosolic transferases. This assumption is supported by the data reported in the literature. For example, Dulik et al. showed that both pentachlorophenyl methyl sulfoxide and pentachlorophenyl methyl sulfone are the substrates for microsomal and cytosolic glutathione-S-transferase of rabbit, monkey, chicken and human liver [36]. According to Campbell et al., both molinate sulfoxide and sulfone are electrophilic and form glutathione conjugates [28].

In the next step, we evaluated the possibility of conjugating glutathione with other atoms of bemethyl. Gunnarsdottir and Elfarra [29] showed that the formation of 6-mercaptopurine (6-MP, structurally similar to metabolite M1 of bemethyl) from PTA was glutathione-dependent and accelerated in the presence of GST. GSTA1-1 was the most active catalyst among the GSTs tested. The authors proposed a mechanism for the formation of 6-MP through the conjugation of glutathione to the double bond of PTA with further release of 6-MP.

According to our data, 2-(vinylthio)benzimidazole is one of the products of bemethyl oxidation (compound **6**, Appendix A). We performed molecular docking of compound **6** into the active site of rGSTA1-1 with further optimization of the complex using the energy minimization method. The result is shown in Figure 10.

According to the data obtained, the distances between the sulfur atom of glutathione and the carbon atoms of the vinyl moiety of the ligand are 4.59 and 4.03 Å for CH_2_= and =CH- groups, respectively. Considering the rather large distances between these atoms and low probability of the formation of compound **6** (score is equal to 0.42, which is ten times lower than for bemethyl sulfoxide, Appendix A), we suppose this pathway for the formation of metabolite M1 is unlikely. Apparently, for the implementation of this mechanism, the presence of oxygen atoms in the structure is also necessary to enhance the reactivity of the attacked atom [29].

Finally, we have aligned the primary sequences of human and rat GSTA1-1 (Figure 11).

Amino acids residues Tyr9, Phe10, Gly14, Arg15, Gln54, Val55, Pro56, Gln67, Thr68, Met107, Phe220 are involved in binding of GSH, bemethyl, BEM-SO, and BEM-SO_2_ in the active center of rGSTA1-1. All of these amino acids, with the exception of Met107, are retained in the structure of human GSTA1-1 (hGSTA1-1). Met107 of rGSTA1-1 is replaced by leucine in the primary sequence of hGSTA1-1. Since it is the methyl group of the methionine residue that plays the main role in the steric interaction with the ligands (Figure 9C,D), similar interaction is expected with one of the methyl groups of leucine. Thus, all the molecular modeling results obtained in rGSTA1-1 can be extrapolated to hSTA1-1. For other GST isoforms, further research is needed.

Summarizing the results of in silico studies (metabolite prediction and molecular docking procedures), we believe that the approach that we have applied allows rapid and effective analysis of the data obtained by LC–MS/HRMS. The 3D-structures of the metabolites can only be unambiguously approved by experimental procedures. However, the presented protocol combining LC–MS/HRMS and in silico studies may be part of comprehensive investigation of xenobiotics metabolism.

## 3. Materials and Methods

### 3.1. Chemicals and Solvents

The object for study was bemethyl pharmaceutical substance FS 42-2525-88 with the main substance content of 97% (by NMR). Deionized water, acetonitrile and formic acid from Thermo Fisher Scientific were used.

### 3.2. Toxicological Experiment

Animal care and use followed the “Rules of Good Laboratory Practice in the Russian Federation”, approved the Ministry of Health of the Russian Federation (Order no. 267 of 19.06.2003).

The rats were acclimatized for three days in Laboratory Animal Service facility at the Research Institute of Hygiene, Occupational Pathology and Human Ecology and had free access to food and water. The experiments were carried out following the animal protocol approved by the Scientific Advisory Board of the Institute. Keeping animals and carrying out manipulations were guided by the rules of humane treatment of animals. (GOST 33216-2014—Guidelines for accommodation and care of animals. Species-specific provisions for laboratory rodents and rabbits.) Male Wistar rats with the average body weight 300 ± 30 g were intragastrically injected with a 1 mL of a 100 mg/mL aqueous solution of bemethyl. Two groups of animals were used in the experiment: an experimental and a control group, five rats in each. The pooled urine from each group of animals was analyzed. Urine samples were collected during the day using metabolic chambers, and were filtered and prepared for LC–MS/HRMS analysis.

### 3.3. Preparation of Urine Samples for LC–MS/HRMS Analysis

Acetonitrile (900 µL) was added to a 300 µL urine sample. The resulting mixture was centrifuged at 14,000 rpm for 5 min, and the supernatant was collected and analyzed by LC–MS/HRMS. This approach excluded the concentration and hydrolysis of the conjugates, which provided the observation of unchanged phase 1 and 2 metabolites.

### 3.4. LC–MS/HRMS Instrumentation and Conditions of Analysis

A Thermo Fisher Scientific Q Exactive hybrid mass spectrometer with an ESI source (Waltham, MA, USA; manufactured by “Thermo Fisher Scientific” (Bremen) GmbH, Germany) and an Xcalibur 4.1 (Thermo Fisher Scientific, Waltham, MA, USA) data acquisition and analysis software was used. Chromatographic separation was performed on an Agilent Zorbax SB-C8 column (150 mm × 4.6 mm; particle size 1.8 µm) under the following eluent gradient conditions: 0 to 1 min; 95% А; 1 to 8 min, 95% to 50% А; 8 to 9 min, 50% to 10% А; 9 to 12 min, 10% А; 12 to 12.1 min, 10% to 95% А; 12.1 to 15 min, 95% А; injected sample 0.01 mL. The mobile phases were (A) 0.1% formic acid in deionized water and (B) 0.1% formic acid in a gradient grade acetonitrile. The eluent flow rate was 0.4 mL/min, and the column temperature was maintained at 35 °C.

Detection was carried out in the high-energy collisional dissociation (HCD) fragmentation mode with normalized collision energy (NCE), monitoring each analyte in a separate mass range with the first mass fixed at *m/z* 50 (Table 3). The minimum resolution was set at 17,500. The characteristic product ions are listed in Table 2. Exact mass measurements were performed under the following conditions: sheath gas flow rate 45 L/h; sheath gas temperature 300 °С; auxiliary gas flow rate 15 L/h; auxiliary gas temperature 380 °С; spray voltage 3500 V. The mass detector was operated in the positive ion mode.

### 3.5. In Silico Analysis of the Metabolism of Bemethyl

The 3D models of the molecules of bemethyl and its metabolites were created and minimized by the method of steepest descent [37] using HyperChem 8.0.8 software package (Hypercube Inc., Gainesville, FL, USA) [38]. The pdb format was exported to a 3D sdf format by means of Online SMILES Translator and Structure File Generator services (Chemical Biology Laboratory, National Cancer Institute, Frederick, MD, USA) [39]. The resulting 3D structures were used for searching the possible metabolites with the BioTransformer open access metabolism prediction software tool [23] and GLORY software tools (University of Vienna, University of Hamburg) [24].

### 3.6. Conjugation of Bemethyl and Its Oxidation Products with Glutathione According to Molecular Modeling Data

Crystal structure of rGSTA1-1A (code 1EV9 [40]) served as a three-dimensional model of the protein. Chain A was used for the molecular docking procedure; other chains and ligand molecules (except for glutathione sulfonate) were removed from the structure. Glutathione sulfonate was manually transformed into the deprotonated form of glutathione by removing oxygen atoms from the pdb file. Then, the missing hydrogens were added with the help of the MGL-Tools 1.5.6 program (The Scripps Research Institute, La Jolla, CA, USA) [41] (glutathione was kept in the deprotonated form) and the mutant amino acids were replaced with native ones according to the primary sequence of rGSTA1-1A using Discovery Studio Visualizer software v16.1.0.15350 (BIOVIA, San Diego, CA, USA) [42]. The tool “Build and Edit Protein” was used, which allows one to automatically build or change an amino acid residue according to the standard topology, then to save a new structure in pdb-format. To resolve clashes, the resulting structure was optimized by energy minimization with the help of GROMACS 2018.1 software (University of Groningen) [43].

Molecular docking of ligands into the binding site of rGSTA1-1 was performed using Autodock Vina 1.1.2 software package (The Scripps Research Institute, La Jolla, CA, USA) [44]. The coordinates of the hydrogen atom of Tyr9 hydroxyl group were used as the center of the search area. The size of search area was set to 20 × 20 × 20 Å^3^. The parameter called ”exhaustiveness”, characterizing the amount of computational effort, was set to 10. The parameter ”energy_range”, characterizing the maximum scatter of energy values of conformations in the output file, was set to 3 kcal/mol. The number of the most optimal conformations in the output file (num_modes) was set to 20. The result of the docking procedure was a set of 20 most probable conformations. The conformations with the minimal distance between the sulfur atom of glutathione and atom C2 of benzimidazole ring were picked up for further analysis. The resulting structures of protein-ligand complexes were optimized by energy minimization in an aqueous solution using GROMACS 2018.1 software [43].

The alignment of the primary sequences of rat and human GSTA1-1 was performed using MultAlin software [45]. The primary sequences were taken from Uniprot Database [46], code P00502 for rGSTA1-1 and P08263 for hGSTA1-1.

## 4. Conclusions

In the present work, we studied the metabolism of bemethyl, detected nine of its urinary metabolites for the first time, and proposed their structural formulas. Suggestions were made on the biotransformation pathways of bemethyl in rats. Two of the most abundant urinary metabolites of bemethyl were detected, M4 and M5 (Figure 8). Metabolite M4 was suggested to result from the hydroxylation and subsequent sulfonating of bemethyl. Metabolite M5 (the first most abundant) is very likely a benzimidazole–acetylcysteine conjugate. In the urine of the rats that received bemethyl in a dose a fortiori that was higher than therapeutic, this metabolite prevailed quantitatively. Therefore this biotransformation pathway is most likely associated with the detoxification of xenobiotics. In general, actoprotectors are widely used in disaster medicine, military medicine, and sports. With this connection, a detailed study of their biotransformation seems to be an urgent task, which is necessary to solve for understanding the molecular mechanisms of their action. These mechanisms, as it is now becoming clear, are far from clearly understood.

The picture of the metabolic transformations of bemethyl, which follows from our analysis, shows that the antioxidant potential of this drug is most likely associated with its consecutive oxidation to sulfoxide and sulfone under the conditions of resistance to oxidative stress. Taking into account that, since 2018, bemethyl has been included in the WADA monitoring program, the study of urinary metabolites and their possible time detection windows is an urgent task, especially given the fact that the unchanged bemethyl is detected in urine in low concentrations and for a short period of time after administration.

## Figures and Tables

**Figure 1 ijms-22-09021-f001:**
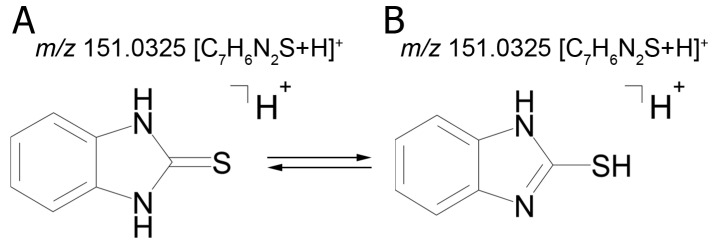
Tautomeric forms of 2-thiobenzimidazole (metabolite M1). (**А**) The thione form of 2-thiobenzimidazole. (**B**) The thiol form of 2-thiobenzimidazole.

**Figure 2 ijms-22-09021-f002:**
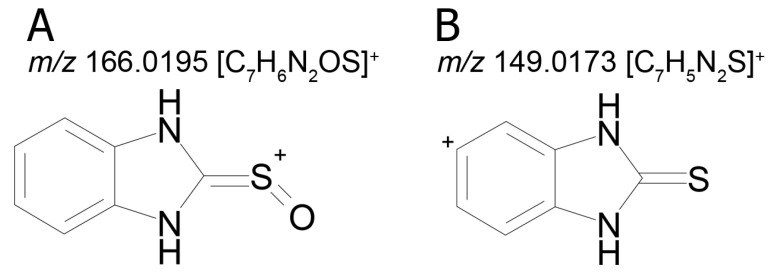
Proposed structures of the product ions of metabolites M2а (**A**) and M2b (**B**).

**Figure 3 ijms-22-09021-f003:**
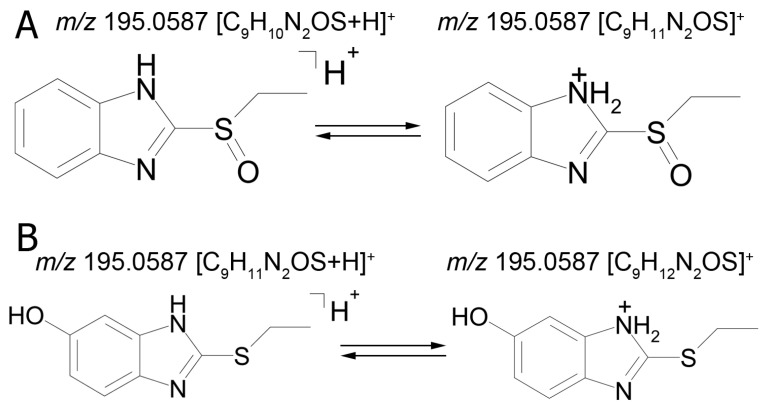
Proposed structural formulas of metabolites M2а (**A**) and M2b (**B**).

**Figure 4 ijms-22-09021-f004:**
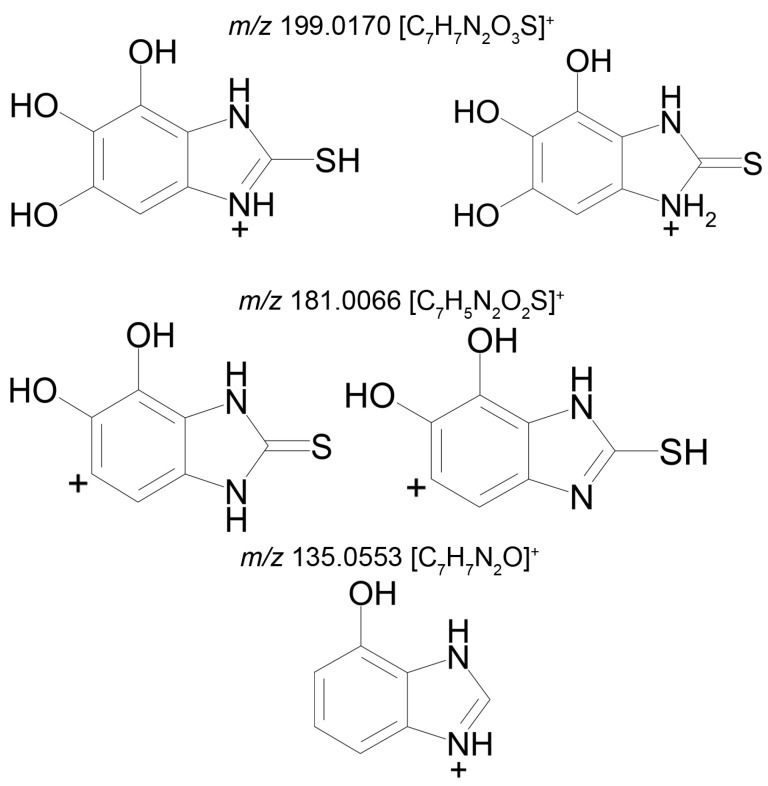
Proposed structures of the product ions of metabolite M3.

**Figure 5 ijms-22-09021-f005:**
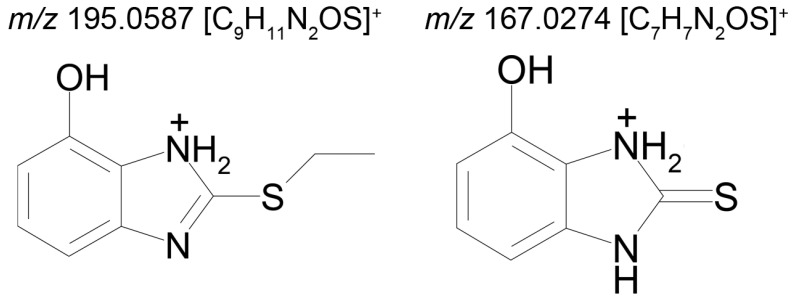
Proposed structures of the product ions of metabolite M4.

**Figure 6 ijms-22-09021-f006:**
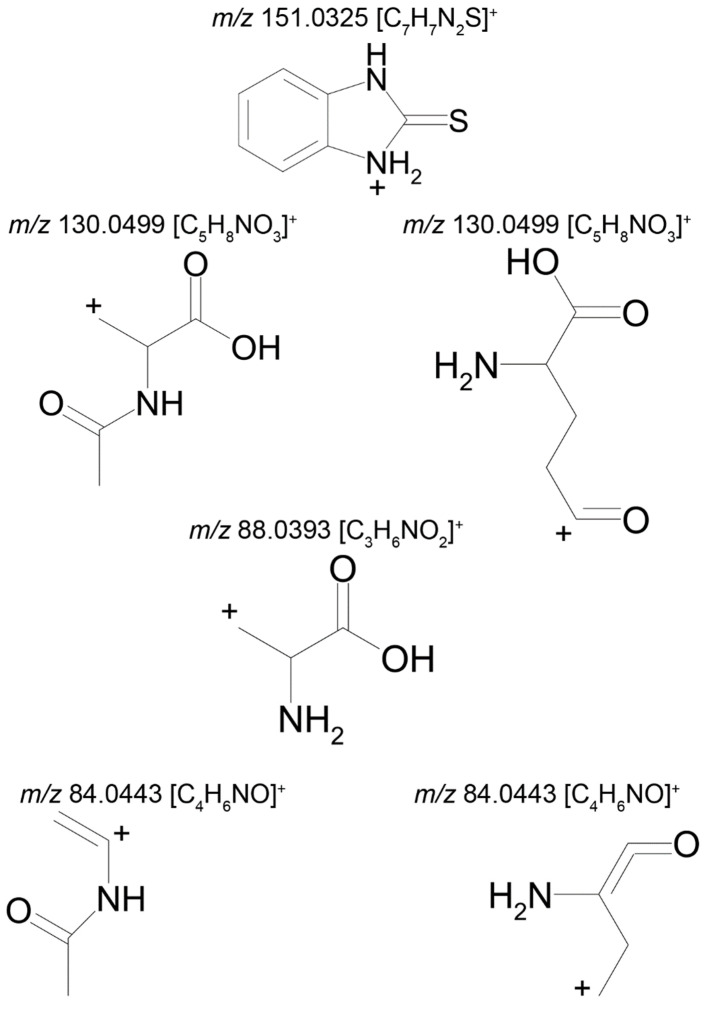
Proposed structures of the product ions of metabolite M5.

**Figure 7 ijms-22-09021-f007:**
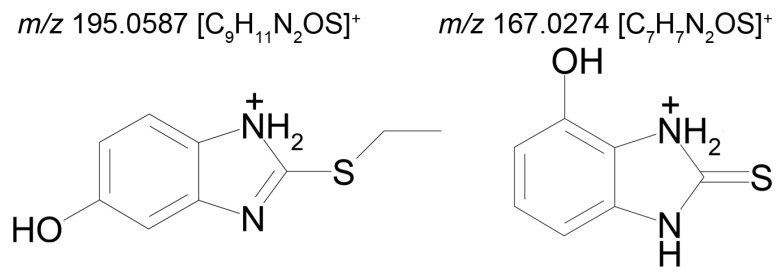
Proposed structures of the product ions of metabolite M6.

**Figure 8 ijms-22-09021-f008:**
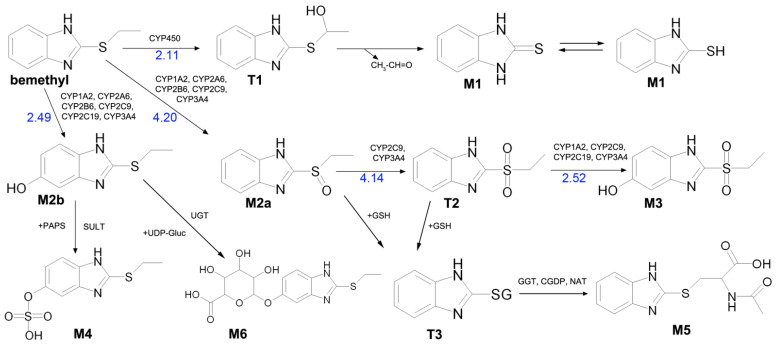
Proposed metabolism pathways of bemethyl. GLORY scores are shown in blue. (CYP) Cytochrome P450 family; (PAPS) 3′-phosphoadenosine-5′-phosphosulfate; (SULT) sulfotransferase; (UDP–Gluc) uridine diphosphate glucuronic acid; (UGT) uridine 5′-diphosphoglucuronosyltransferase; (GSH) glutathione; (GGT) γ-glutamyltransferase; (CGDP) cysteinylglycine dipeptidase; (NAT) *N*-acetyltransferase. (T1, T2, and T3) Transitional metabolites of bemethyl and (M1, M2a, M2b, M3, M4, M5, and M6) final metabolites of bemethyl.

**Figure 9 ijms-22-09021-f009:**
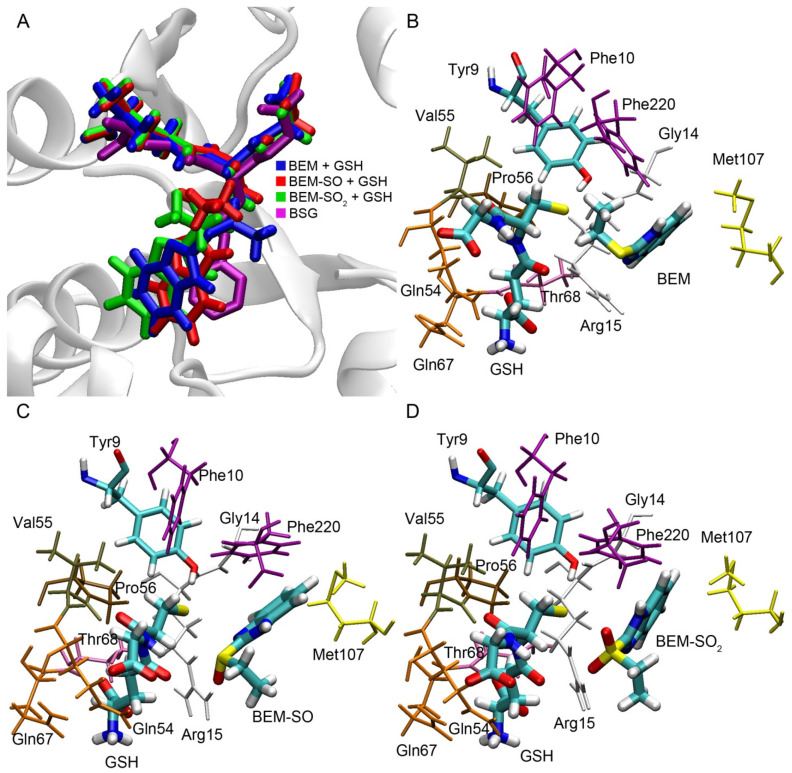
Bemethyl and its derivatives bound inside the ligand-binding site of the rat glutathione S-transferase alpha 1 (rGSTA1-1) according to molecular docking procedure. (**A**) Superposition of docked bemethyl and its derivatives with the S-benzylglutathione complex in the active center of human glutathione S-transferase alpha 1 (structure code 1GUH); (**B**) Bemethyl inside the rGSTA1-1 binding site; (**C**) Bemethyl sulfoxide inside the rGSTA1-1 binding site; (**D**) bemethyl sulfone inside the rGSTA1-1 binding site. BEM, bemethyl; BEM-SO, bemethyl sulfoxide; BEM-SO_2_, bemethyl sulfone; BSG, S-benzyl-glutathione; GSH, glutathione.

**Figure 10 ijms-22-09021-f010:**
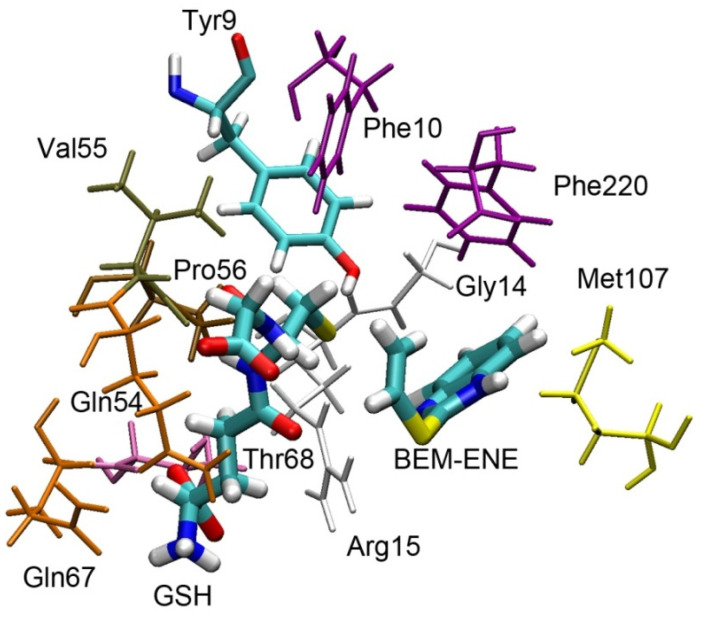
2-(Vinylthio)benzimidazole bound inside the ligand-binding site of the rat glutathione S-transferase alpha 1 (rGSTA1-1) according to molecular docking procedure. BEM-ENE, 2-(vinylthio)benzimidazole; GSH, glutathione.

**Figure 11 ijms-22-09021-f011:**
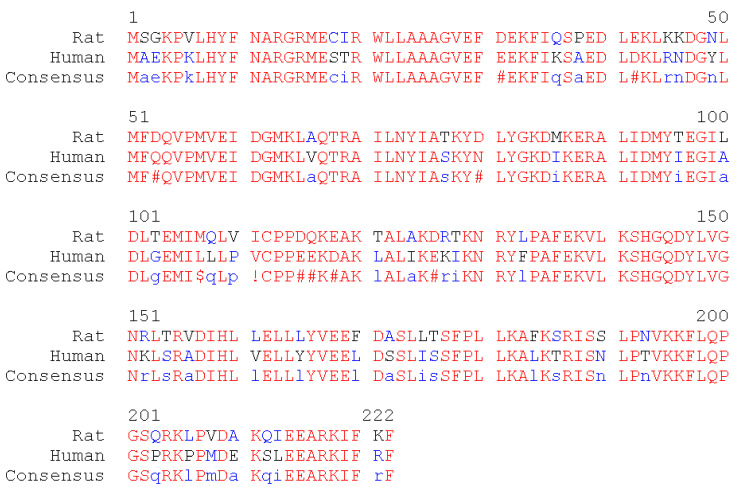
Alignment of primary sequences of rat and human glutathione S-transferase A1. Red and blue letters indicate the amino acids with high and low homology in different organisms, respectively. Black letters show neutral amino acid substitutions. The bottom line is the so-called consensus sequence, which is the generalized sequence that is obtained by comparing letters in the alignment columns. When homology is high, the consensus line represents the predominant amino acid, indicated by a red capital letter. When homology is low, the consensus line represents the predominant amino acid indicated by a lowercase blue letter. The symbol # is displayed in the consensus sequence when a polar amino acids is replaced by a polar one (glutamine, glutamate, asparagine, aspartate); ! means that valine is replaced by isoleucine; $ indicates that leucine is replaced by methionine.

**Table 1 ijms-22-09021-t001:** Molecular and structural formulas of bemethyl and its analogs.

Name	Molecular Formula	Structural Formula
Bemethyl	C_9_H_10_N_2_S	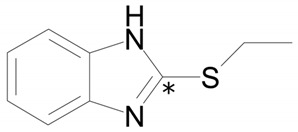
Triclabendazole	C_14_H_9_Cl_3_N_2_OS	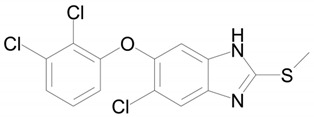
Afobazole (Fabomotizole)	C_15_H_21_N_3_O_2_S	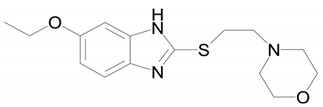
Pantoprazole	C_16_H_15_F_2_N_3_O_4_S	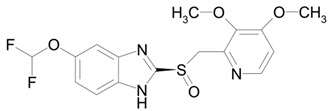
Omeprazole	C_17_H_19_N_3_O_3_S	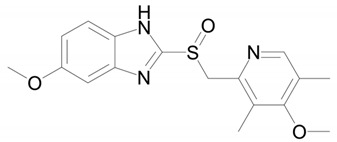

*—С2 position of the benzimidazole ring.

**Table 2 ijms-22-09021-t002:** Principal chromatographic and tandem mass spectral characteristics of bemethyl and its metabolites.

Compound	Proposed Structure	RT, min	Precursor Ion		Product Ion	
			*m/z*	Molecular Formula	*m/z*	Molecular Formula
Bemethyl	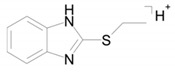	10.7	179.0638	[C_9_H_10_N_2_S+H]^+^	151.0324 93.0581 118.0531	[C_7_H_6_N_2_S]^+^ [C_6_H_7_N]^+^ [C_7_H_6_N_2_]^+^
M1а M1b	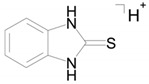 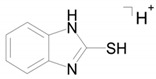	9.40 11.01	151.0325	[C_7_H_6_N_2_S+H]^+^ [C_7_H_6_N_2_S+H]^+^	92.04948 93.0581 118.0531 119.0604	[C_6_H_6_N]^+^ [C_6_H_7_N]^+^ [C_7_H_6_N_2_]^+^ [C_7_H_7_N_2_]^+^
M2а M2b *	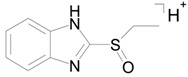 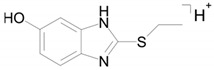	9.13 11.42	195.0587	[C_9_H_10_N_2_OS+H]^+^ [C_9_H_10_N_2_OS+H]^+^	134.0483 166.0195 167.0278 149.0168 167.0279	[C_7_H_6_N_2_O]^+^ [C_7_H_6_N_2_OS]^+^ [C_7_H_7_N_2_OS]^+^ [C_7_H_5_N_2_S]^+^ [C_7_H_7_N_2_OS]^+^
M3a * M3b *	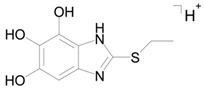 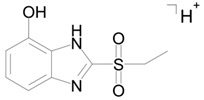	9.26 10.51	227.0485	[C_9_H_10_N_2_O_3_S+H]^+^	199.0170 181.0066 153.0117 135.0553	[C_7_H_7_N_2_O_3_S]^+^ [C_7_H_5_N_2_O_2_S]^+^ [C_6_H_5_N_2_OS^+^] ** [C_7_H_7_N_2_O]^+^
M4 *	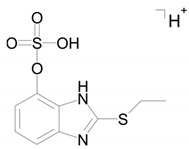	9.14	275.0155	[C_9_H_10_N_2_O_4_S_2_+H]^+^	195.0587 167.0274	[C_9_H_11_N_2_OS]^+^ [C_7_H_7_N_2_OS]^+^
M5a M5b	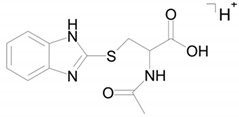 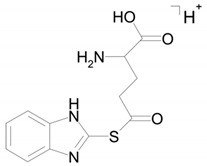	9.37	280.0689	[C_12_H_13_N_3_O_3_S+H]^+^	151.0325 130.0499 88.0393 84.0443	[C_7_H_7_N_2_S]^+^ [C_5_H_8_NО_3_]^+^ [C_3_H_6_NО_2_]^+^ [C4H_6_NО]^+^
M6 *	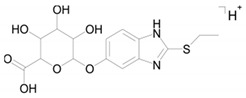	8.21	371.0908	[C_15_H_18_N_2_O_7_S+H]^+^	195.0587 167.0274	[C_9_H_11_N_2_OS]^+^ [C_7_H_7_N_2_OS]^+^

*—The position of the OH group is indicated arbitrarily; **— Molecular formula is suggested

**Table 3 ijms-22-09021-t003:** Detection conditions of bemethyl and its metabolites.

Compound	Precursor Ion, *m/z*	Scan Range, *m/z*	NCE, %
Bemethyl	179.0638	50–200	35
M1 (a, b)	151.0325	50–170	55
M2 (a, b)	195.0587	50–215	35
M3 (a, b)	227.0485	50–250	35
M4	275.0155	50–300	35
M5 (a, b)	280.0689	50–305	35
M6	371.0908	50–395	35

NCE, normalized collision energy.

## Data Availability

The data presented in this study are available on request.

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
