# Peer review of "Investigation of Bemethyl Biotransformation Pathways by Combination of LC–MS/HRMS and In Silico Methods"

_ijms, 2021, doi:10.3390/ijms22169021_

Round 1

Reviewer 1 Report

The authors describe an interesting investigation of Bemethyl biotransformation pathways using (1) LC-HRMS(/MS) to elucidate the possible metabolites formed in rat urine 24h after bemethyl administration and (2) in silico approaches to link the biotransformation products and their corresponding metabolic pathways. The paper is globally well written but there are still some inaccuracies and rewordings to be done in the part "2.1. Mass spectral identification...". The described study is well conducted and the results are well presented. Nevertheless, it would be appreciated to present, in the supplementary data, the recorded tandem mass spectra of the different metabolites to support structural identification.

I am in favor of the publication of this paper in Int. J. Mol. Sci. provided a number of clarifications are made according to the following recommendations:

  • In the title, LC-HRMS/MS would advantageously replace LC-MS/HRMS;
  • The chemical structures in Figure 8 are well presented. Please use the same style/software for the drawing of every other chemical structures in the document (including figures 1 to 7 and tables 1 and 2). In particular, in Table 1 and Table 2, authors should pay attention to the size of the structures because some of them are truncated. In table 2, some charges are missing (not visible) in the structures of the positive ions.
  • Page 3, there is a repetition of the same sentences line 76-80 and line 103-106. Please modify.
  • The term “mass numbers” (for instance p.3 line 98 and 99) is used improperly and should be replace with “accurate masses”. Please modify.
  • Throughout the document, when referring to MS/MS spectra, please add "tandem" before "mass spectra". For example, P5, line 115, it should read "Thus, the tandem mass spectra of peaks..." instead of "Thus, the mass spectra of peaks...". Please make the necessary changes.
  • In Table 2 and throughout the document, the accurate masses (m/z) of precursor and fragment ions must be written in a homogeneous way with 4 digits after the decimal point. Please correct.
  • 5, line114-115: isomers have, by definition, the same molecular formula. Please correct the sentence. M1a and M1b may be protomers or tautomers.
  • Figure 1 is not really informative. The presentation of the equilibrium between the thione-thiol is all right, but part (C) is incorrect: the arrow should be a double arrow like in Figure 8 for M1, and both structures are ions, therefore there is no need to add a proton on the arrow. Idem for Figure 3. Please modify. Note that the charge/proton could be located at any heteroatom position.
  • Most of the metabolite structures presented are hypothetical structures. Therefore, please add "proposed structures" wherever appropriate. For example, the caption for Figure 2 should be "proposed structures of the product ions of metabolite M2a and M2b".
  • 6, line148-150: please rephrase the sentence that is not understandable.
  • The authors mentioned the possible formation of hydroxylation product (e.g. M2b and M3a) or S-oxidation products (e.g. M2a and M3b). It is well known (since the 90s) that in metabolic investigation studies, hydroxylation and S-oxidation products may be differentiated by hydrogen/deuterium exchange (HDX) experiments. It would have be valuable to perform such experiment to confirm or reject some of the potential metabolite structures. At least the authors should mention the possibility of doing so.
  • In Table 2, a star (indicating the arbitrary –O position in the aromatic ring) is missing for M4 and M6.
  • Still in Table 2 and in the corresponding text p.6 lin153-159, two peaks are detected with the same m/z for M3. The authors mentioned two possible structures: a tri-hydroxylated compound or a mono-hydroxylated sulfone compound. Both compounds are isomers with different chemical functions (functional isomers) and should be differentiated based on their tandem mass spectra. Nevertheless, the authors specify that both peaks (M3a and M3b) display identical tandem mass spectra. Therefore, M3a and M3b are most probably positional isomers rather than functional isomers (e.g. sulfone with 1 aromatic hydroxyl at different position). Please clarify/modify.
  • 8, line 174: involves should be replace by “could provide”.

P.8, line 189: 1 digit after the decimal point is enough for low-resolution m/z values. On the contrary, line 190, 4 digits after the decimal point are required for high-resolution m/z values. Please modify.

Author Response

We are grateful to Reviewer-1 for the valuable comments on the manuscript. We have substantially modified the manuscript to address the points raised. The changes are detailed below and highlighted in the revised and improved manuscript.

  • Nevertheless, it would be appreciated to present, in the supplementary data, the recorded tandem mass spectra of the different metabolites to support structural identification.

The tandem mass spectra have been added to Supplementary. Numeration of supplementary figures has been changed.

  • In the title, LC-HRMS/MS would advantageously replace LC-MS/HRMS;

corrected

  • The chemical structures in Figure 8 are well presented. Please use the same style/software for the drawing of every other chemical structures in the document (including figures 1 to 7 and tables 1 and 2). In particular, in Table 1 and Table 2, authors should pay attention to the size of the structures because some of them are truncated. In table 2, some charges are missing (not visible) in the structures of the positive ions.

corrected.

  • Page 3, there is a repetition of the same sentences line 76-80 and line 103-106. Please modify.

corrected

  • The term “mass numbers” (for instance p.3 line 98 and 99) is used improperly and should be replace with “accurate masses”. Please modify.

corrected

  • Throughout the document, when referring to MS/MS spectra, please add "tandem" before "mass spectra". For example, P5, line 115, it should read "Thus, the tandem mass spectra of peaks..." instead of "Thus, the mass spectra of peaks...". Please make the necessary changes.

corrected

  • In Table 2 and throughout the document, the accurate masses (m/z) of precursor and fragment ions must be written in a homogeneous way with 4 digits after the decimal point. Please correct.

corrected

  • 5, line114-115: isomers have, by definition, the same molecular formula. Please correct the sentence. M1a and M1b may be protomers or tautomers.

corrected for tautomers

  • Figure 1 is not really informative. The presentation of the equilibrium between the thione-thiol is all right, but part (C) is incorrect: the arrow should be a double arrow like in Figure 8 for M1, and both structures are ions, therefore there is no need to add a proton on the arrow. Idem for Figure 3. Please modify. Note that the charge/proton could be located at any heteroatom position.

The recommended corrections are made. Due to the fact that the exact localization of the ion charge in most cases is impossible to establish, we took the "+" sign out of brackets. The note has been added to the 3d paragraph of page 6: The localization of the OH group may be different, as well as charge/proton could be located at any heteroatom position (here and below).

  • Most of the metabolite structures presented are hypothetical structures. Therefore, please add "proposed structures" wherever appropriate. For example, the caption for Figure 2 should be "proposed structures of the product ions of metabolite M2a and M2b".

corrected

  • 6, line148-150: please rephrase the sentence that is not understandable.

New version: Figure 3 represents the proposed structural formulas of metabolites M2. The lo-calization of the OH group may be different., as well as charge/proton could be located at any heteroatom position (here and below).

  • The authors mentioned the possible formation of hydroxylation product (e.g. M2b and M3a) or S-oxidation products (e.g. M2a and M3b). It is well known (since the 90s) that in metabolic investigation studies, hydroxylation and S-oxidation products may be differentiated by hydrogen/deuterium exchange (HDX) experiments. It would have be valuable to perform such experiment to confirm or reject some of the potential metabolite structures. At least the authors should mention the possibility of doing so.

Completely agree.  Added a recommended comment : It is well known (since 1990s) that, in metabolic studies, hydroxylation and S-oxidation products can be differentiated by hydrogen/deuterium exchange (HDX) experiments. Investigation of bemethyl metabolites by HDX technique is one of our future research challenges.

  • In Table 2, a star (indicating the arbitrary –O position in the aromatic ring) is missing for M4 and M6.

Corrected

  • Still in Table 2 and in the corresponding text p.6 lin153-159, two peaks are detected with the same m/z for M3. The authors mentioned two possible structures: a tri-hydroxylated compound or a mono-hydroxylated sulfone compound. Both compounds are isomers with different chemical functions (functional isomers) and should be differentiated based on their tandem mass spectra. Nevertheless, the authors specify that both peaks (M3a and M3b) display identical tandem mass spectra. Therefore, M3a and M3b are most probably positional isomers rather than functional isomers (e.g. sulfone with 1 aromatic hydroxyl at different position). Please clarify/modify.

Modified. Two variants of the proposed structures of the isomers of M3 are presented in Table 2, and the proposed structures of the product ions, generated by the Mass Frontier software, are shown in Figure 4.

  • 8, line 174: involves should be replace by “could provide”.

Modified: This metabolite was not identified by the MetWorks software, because none of the metabolic pathways entered in the program involve benzimidazole ring opening.

P.8, line 189: 1 digit after the decimal point is enough for low-resolution m/z values. On the contrary, line 190, 4 digits after the decimal point are required for high-resolution m/z values. Please modify.

The m/z values of the precursor and most characteristic product ions in the low-resolution mode are 355.00 and 179.00, respectively, so quoted in the cited source [9, Kwiatkowska et al.]

Reviewer 2 Report

This paper looks sound, and is of interest to the journal’s readership. The work is very interesting. However, there are some issues that need to be further explored by the authors, and therefore the paper needs to be improved before publication in the International Journal of Molecular Sciences can be considered. I think it is acceptable after some modifications.

I have some comments/questions to the authors:

- Introduction:  The authors should include information about this compound and its use in sports and the importance of its monitoring according to the WADA.

- Please change HPLC-MS/MS to LC–MS/HRMS or LC–MS/MS throughout the manuscript.

-Experimental: Please include a justification on why a concentration of 100 mg/mL was chosen.
What was the number of animals used?

- Please include a chromatogram of bemethyl and one of a blank urine sample. Did you evaluate matrix effects? Please include more information about LC-MS/MS conditions, e.g. volume of injection, scan range of m/z and resolving power. Did the authors perform some sort of enzymatic hydrolysis?

- Kwiatkowska et al. (https://doi.org/10.1002/dta.2524) recommends the use of liquid-liquid extraction method for a confirmatory analysis. In the authors’ opinion, is a previous extraction recommended or, conversely, is a direct injection sufficient to detect the compound and its metabolite?

- In the supplementary files, more information about the GLORY results should be included as well the pharmacokinetic properties of bemethyl.

- Figure 8, perhaps it would be easier to the reader to see the GLORY scores together with each predicted metabolite.

Author Response

We are grateful to Reviewer-2 for the valuable comments on the manuscript. We have substantially modified the manuscript to address the points raised. The changes are detailed below and highlighted in the revised and improved manuscript.

- Introduction:  The authors should include information about this compound and its use in sports and the importance of its monitoring according to the WADA.

A phrase has been added to the introduction: Since 2018, bemethyl is included to the Monitoring Program of the World Anti-Doping Agency, which highlights the challenge of identifying its urinary metabolites.

- Please change HPLC-MS/MS to LC–MS/HRMS or LC–MS/MS throughout the manuscript

HPLC-MS/MS corrected throughout to LC–MS/HRMS

-Experimental: Please include a justification on why a concentration of 100 mg/mL was chosen.
What was the number of animals used?

An explanation of the choice of dose is given in the second paragraph of the section “Results and discussion”: The extremely high dose of bemethyl was chosen with two goals in mind: 1) to detect as many major metabolites as possible and preserve the original component ratio of the sample, because concentration inevitably entails a change in its composition and 2) to identify metabolites associated with detoxification.

 - Please include a chromatogram of bemethyl and one of a blank urine sample. Did you evaluate matrix effects? Please include more information about LC-MS/MS conditions, e.g. volume of injection, scan range of m/z and resolving power. Did the authors perform some sort of enzymatic hydrolysis?

Chromatograms of bemethyl, its metabolites in urine of exposed rats, and blank urine are now presented in Supplementary materials. Numeration of Supplementary figures has been changed.

Section 3.4. "LC-MS/HRMS instrumentation and conditions of analysis" is supplemented by relevant data:

A Thermo Fisher Scientific Q Exactive hybrid mass spectrometer with ESI source and an Xcalibur data acquisition and analysis software was used. … injected sample 0.01 mL.

Detection was carried out in the high-energy collisional dissociation (HCD) fragmentation mode with normalized collision energy (NCE), monitoring each analyte in a separate mass range with the first mass fixed at m/z 50 (Table 3 in attachment). The minimum resolution was set at 17,500. The characteristic product ions are listed in Table 2.

  1. Results and discussion, 3rd paragraph: Enzymatic hydrolysis was not used for the direct determination of conjugates.

- Kwiatkowska et al. (https://doi.org/10.1002/dta.2524) recommends the use of liquid-liquid extraction method for a confirmatory analysis. In the authors’ opinion, is a previous extraction recommended or, conversely, is a direct injection sufficient to detect the compound and its metabolite?

  1. Results and discussion, 3rd paragraph: Extraction procedures will be developed for streaming target analysis. During the primary identification phase, direct urine analysis seems to be optimal.

- In the supplementary files, more information about the GLORY results should be included as well the pharmacokinetic properties of bemethyl.

The detailed information about the GLORY procedure and its results has been added to Supplementary materials

The study of the pharmacokinetics of metabolites was not included into the objectives of the study. We have added a reference to literature at the beginning of the third paragraph of Introduction :

“Even though the therapeutic effectiveness of bemethyl is well documented [4,5], as well as its pharmacokinetics [6,7], there is a significant gap in knowledge regarding its metabolism products and their quantitative and qualitative characteristics.”

A brief description of bemethyl pharmacokinetics obtained by the authors of [6,7] is given in the following paragraphs.

- Figure 8, perhaps it would be easier to the reader to see the GLORY scores together with each predicted metabolite.

Figure 8 has been revised according to the comments.

Reviewer 3 Report

The paper deals with a metabolite study of the derivatives produced in a rat model aimed at understanding the pharmacokinetics and pharmacodynamics of Bemethyl  an actoprotector,  antihypoxant and moderate psychostimulant drug. The work has been performed using a combination of liquid chromatography-high-resolution mass spectrometry and in silico studies. A wealth of information is provided, including the identification of several bemethyl metabolites and the molecular docking of bemethyl and its derivatives to the binding site of glutathione S-transferase that revealed the possible mechanism of bemethyl conjugation with glutathione.

On the other hand, the manuscript somewhere lacks of clarity and in my opinion requires a revision in order to improve the final outcome and the easy understanding by the general reader. In particular, the Result and Discussion section should start with a clear description of the used experimental animal model possibly describing the modification (if any) introduced with respect to the model used in previous work (ref.9). Therefore, as the Author state at the beginning of the section, they should describe and discuss the results of the HPLC–MS/MS analysis also supported by literature data on the metabolic transformations of other drugs of the benzimidazole.

In this respect, it seems that the authors also for the description of the HPLC–MS/MS analysis rely on their previous work without giving further details on the obtained chromatogram, the relative abundance of the focused peaks among them and with respect to other urine metabolites. “In our previous work, in  the mass spectra of rat urine samples we identified six compounds that could be considered as possible bemethyl metabolites [9]. In the present work we performed an in-depth study of the composition of the products of bemethyl metabolism in rats and made assumptions about the metabolic pathways. The principal mass spectral and chromatographic characteristics of bemethyl and its metabolites are listed in Table 2.”

In conclusion. The paper requires the introduction of a thorough description and discussion of the animal experimental model used and the chromatographic and ”primary mass spectral information” acquisition besides the MetWorks software description. The usefulness of this latter could be then much better described by the authors and understood by the reader.

Author Response

We are grateful to Reviewer-3 for the valuable comments on the manuscript. We have substantially modifiede manuscript to address the points raised. The changes are detailed below and highlighted in the revised and improved manuscript.

In particular, the Result and Discussion section should start with a clear description of the used experimental animal model possibly describing the modification (if any) introduced with respect to the model used in previous work (ref.9). Therefore, as the Author state at the beginning of the section, they should describe and discuss the results of the HPLC–MS/MS analysis also supported by literature data on the metabolic transformations of other drugs of the benzimidazole.

2. Results and discussion section, paragraph 1 is supplemented as follows:

With all the great potential of LC-MS/HRMS to identify xenobiotic metabolites in multicomponent biomatrices, the xenobiotic biotransformation pathways are impossible to establish without additional information. Such information can be gained from molecular modeling analysis, as well as known data on the metabolism of the closest structural analogs of the studied xenobiotics. Molecular modeling allows the prediction of possible biotransformation pathways with an estimate of the probability of each possible pathway. Published data on the biotransformation of structural analogs provide the necessary supporting information. Thus, the biotransformation pathway of bemethyl to sulfoxide and sulfone, with a stable hydroxylated form arising only with sulfoxide but not with sulfone, reveals a complete analogy with the biotransformation of triclabendazole. The ability of pantoprazole and omeprazole to form a conjugate with N-acetylcysteine suggests the same biotransformation pathway of bemethyl.

In this respect, it seems that the authors also for the description of the HPLC–MS/MS analysis rely on their previous work without giving further details on the obtained chromatogram, the relative abundance of the focused peaks among them and with respect to other urine metabolites. “In our previous work, in the mass spectra of rat urine samples we identified six compounds that could be considered as possible bemethyl metabolites [9]. In the present work we performed an in-depth study of the composition of the products of bemethyl metabolism in rats and made assumptions about the metabolic pathways. The principal mass spectral and chromatographic characteristics of bemethyl and its metabolites are listed in Table 2.”

The tandem mass spectra and chromatograms have been added to Supplementary. Numeration of supplementary figures has been changed.

In conclusion. The paper requires the introduction of a thorough description and discussion of the animal experimental model used and the chromatographic and ”primary mass spectral information” acquisition besides the MetWorks software description. The usefulness of this latter could be then much better described by the authors and understood by the reader.

Primary data from HPLC-MS/MS analysis, including mass chromatograms and mass spectra illustrating the identification of bemethyl and its metabolites, are presented in Supplementary materials.

1. Introduction, the last paragraph: In [9], we used a different experimental model to study bemethyl metabolism: a course of intragastric administration at a daily dose of 25 mg/kg. This model is physiologically more adequate but less suitable for identifying minor metabolites. In the present work, using a single dose of 330 mg/kg intragastrically, we have performed for the first time an in-depth study of the composition of the products of bemethyl metabolism in rats

2. Results and discussion, the 2-d paragraph: The animal experimental model used would be unsuitable for pharmacokinetic studies and only serves to identify metabolites.

Round 2

Reviewer 2 Report

The authors have adequately responded to all my recommendations and the quality of the manuscript has been considerably improved.

Reviewer 3 Report

The Authors have essentially dealt with the reviewer suggestions/questions